# Understanding the implementation and adoption of an information technology intervention to support medicine optimisation in primary care: qualitative study using strong structuration theory

Mark Jeffries,[1,2] Denham Phipps,[1,2] Rachel L Howard,[3] Anthony Avery,[4] Sarah Rodgers,[4] Darren Ashcroft[1,2]

► Prepublication history and additional material are available. To view these files please visit the journal online (http://dx.doi.org/ 10.1136/bmjopen-2016-014810).

[1]Centre for Pharmacoepidemiology and Drug Safety, School of Health Sciences, University of Manchester, Manchester, UK
[2]NIHR Greater Manchester Primary Care Patient Safety Translational Research Centre, University of Manchester, Manchester, UK
[3]Research Pharmacist, Ryde, Isle of Wight, UK
[4]Division of Primary Care, University of Nottingham, Nottingham, UK

**Correspondence to**
Mark Jeffries;
mark.jeffries@manchester.ac.uk

## ABSTRACT

**Objectives** Using strong structuration theory, we aimed to understand the adoption and implementation of an electronic clinical audit and feedback tool to support medicine optimisation for patients in primary care.

**Design** This is a qualitative study informed by strong structuration theory. The analysis was thematic, using a template approach. An a priori set of thematic codes, based on strong structuration theory, was developed from the literature and applied to the transcripts. The coding template was then modified through successive readings of the data.

**Setting** Clinical commissioning group in the south of England.

**Participants** Four focus groups and five semi-structured interviews were conducted with 18 participants purposively sampled from a range of stakeholder groups (general practitioners, pharmacists, patients and commissioners).

**Results** Using the system could lead to improved medication safety, but use was determined by broad institutional contexts; by the perceptions, dispositions and skills of users; and by the structures embedded within the technology. These included perceptions of the system as new and requiring technical competence and skill; the adoption of the system for information gathering; and interactions and relationships that involved individual, shared or collective use. The dynamics between these external, internal and technological structures affected the adoption and implementation of the system.

**Conclusions** Successful implementation of information technology interventions for medicine optimisation will depend on a combination of the infrastructure within primary care, social structures embedded in the technology and the conventions, norms and dispositions of those utilising it. Future interventions, using electronic audit and feedback tools to improve medication safety, should consider the complexity of the social and organisational contexts and how internal and external structures can affect the use of the technology in order to support effective implementation.

## INTRODUCTION

Prescribing medicines to patients is the most common clinical intervention in primary

### Strengths and limitations of this study

► This is the first study to explore the implementation of electronic audit and feedback systems to improve medication safety in primary care using strong structuration theory.
► Strong structuration theory was found to be particularly valuable for unpicking why the system was used and the different motivations, ambitions, aims and attitudes of a range of stakeholders.
► This was an exploratory study that relied mainly on interview and focus group data from a number of key stakeholders located in one clinical commissioning group in England.
► Additional insights may have been gained by undertaking ethnographic observation to discover exactly the ways people utilised the electronic medicines optimisation system (EMOS).

care. However, with high volumes of medicines prescribed in primary care,[1] the prevalence of repeat prescribing and the increased burden and complexity of multimorbidity and related polypharmacy,[2 3] there is an increased likelihood that prescribing or monitoring errors can occur.[4 5] Recent studies using prescribing safety indicators to investigate the prevalence of hazardous prescribing in primary care found 5.2%–5.5% of patients to be at risk of potentially hazardous prescribing and 7.6%–11.8% of patients not receiving recommended monitoring tests.[6 7] Some medication errors may not lead to harm; however, approximately 13% of patients have experienced an adverse drug event after receiving prescription medication in primary care, and many of those have been serious enough for patients to seek medical assistance at hospital.[8 9] The monitoring of patients in receipt of prescription

medication is therefore considered important in order to avoid potentially serious adverse drug events.

In the UK, national and local policies have set out recommendations for medication safety improvement.[10–12] 'An Organisation with a Memory'[13] set out the necessity for the establishment of a patient safety culture within healthcare organisations. This emphasised the importance of organisational practices. Policy has also set out how the utilisation of information technology (IT) presents opportunities to fulfil medication safety requirements and that the contribution of information systems should be maximised.[11] This was further enhanced by Department of Health recommendations in the report 'Building a Safer NHS for Patients: Improving Medication Safety',[10] which recommended that steps to safer prescribing may include the implementation of effective IT systems particularly those systems that might highlight and give warnings to medical staff of prescription errors. Similar recommendations have suggested there is a need to develop systems that optimise the use of medicines and that this might include improved electronic decision support for clinicians.[12] Locally, the clinical commissioning group (CCG) that formed the setting for this study operated a prescribing incentive scheme designed to improve the quality of prescribing, to respond to the requirements of national guidelines and to reduce excessive prescribing and costs, which was incentivised by small financial rewards for general practices.[14]

Healthcare IT systems may be useful for monitoring medication usage. However, the implementation of such IT has not always been successful, with technology being resisted, not used effectively or used differently than was planned. Previous research has suggested that reasons for this might reside in the design and functionality of the technology. Poorly designed or implemented IT systems have been seen to create cognitive overload[15] and disrupt workflow.[16] Furthermore, IT systems may be utilised in ways unintended by developers, either to overcome problems with design or as new uses for the technology become apparent.[17–19] However, such tailoring of systems suggests a dynamic where implementation actually involves interpretation and adaptation of systems to fit existing work practices or changes to work practices in order to adapt to the new system.[20–25] In other words, the success or failure of an IT implementation could be seen as being shaped by interactions between the technology, the users and the social and organisational processes.[26–28] This sociotechnical view, rather than focusing only on the functionality of systems, takes into account the complex nature of healthcare and the cultural, social and organisational aspects of the workplaces.[22–29]

Strong structuration theory (SST) has been proposed as a way of examining these sociotechnical aspects of healthcare IT implementation.[30] It is based on Giddens' structuration theory, which proposed a relationship between structures (such as social norms, political and economic institutions) and agency (people's actions and choices).[31] According to Stones,[30] SST extends this structure–agency relationship to include the following elements (see figure 1):[32]

▶ *External structures*, which are the physical social or economic context in which action is contemplated. External structures are built through social positions, practices and networks of social

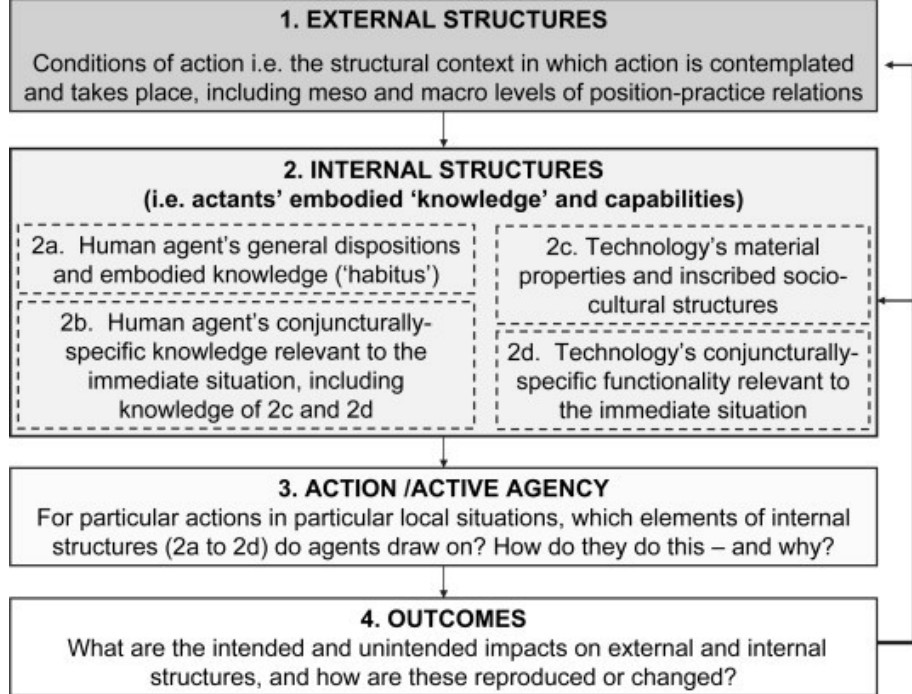

**Figure 1** Strong structuration theory incorporating a technology dimension (adapted from Stones[30]), from Greenhalgh and Stones [32]

relationships.[29][32] These could include hierarchical relationships between employers and employees, professional roles, local and national guidelines, governance measures, regulations, professional codes of practice, as well as local work practices and interactions among groups of stakeholders.[30][32]

► *Internal structures,* which are manifested in two ways: first, as the skills, dispositions, ambitions, attitudes, values, past experiences of actors and ways of viewing the world; second, as the actors' knowledge of rules, conventions, obligations and social norms, which may involve partial understandings and past experiences. These inform how one is supposed to act in specific situations in the here and now, based on the agents understanding of external structures.[30][32][33]

► *Agency,* which is how and why agents draw on internal structures to act in particular ways in specific situations.[32]

► *Outcome,* which is the way agency affects external or internal structures and how they are maintained or changed.[32]

Stones and Greenhalgh[32] further explained the role of technology in SST: rather than there being symmetry between technology and human actors, they are instead separate and may act in different ways.[32] Technology incorporates procedures, codes, material properties and standards that can enable or constrain use[19][32–34]; it is therefore seen as shaping human actions by making certain actions possible.[35] Previous studies[29][34] suggest that SST can illuminate the implementation and adoption of IT by understanding how people 'take action with respect to technologies'; in other words, what people actually do with the systems and to what effect.[32] SST has been previously used to understand the ways a large-scale healthcare IT intervention, designed to assist patients and general practitioners (GPs) to book hospital outpatient appointments, was resisted or adopted.[34]

This study uses SST to examine a new electronic medicines optimisation system (EMOS)[36] that was implemented in a primary care locality. The EMOS allows different stakeholders—GPs, CCG managers, pharmacists, general practice managers and patients—access to real-time anonymised patient data, including medical diagnoses, prescribed medications and laboratory test results. It comprises a secure patient database and a web-based user interface that extracts patient-specific data from the general practice clinical record system. The interface provides a number of user functions; these include reviewing a specific patient health record, identifying patients who are at risk of a medication-related adverse event, such as those who are on inappropriate combinations of drugs or those who have not received appropriate monitoring and carrying out clinical audits on a subset of patients.[36] The EMOS also allowed clinicians and managers in the health locality to audit prescribing practices across general practices and make comparisons against national guidelines. Patients have access to the system through a patient passport, which allows them to view their medications and test results. In this context, it was felt that SST would unpick the ways in which users of the system drew on their dispositions, attitudes, skills and ambitions and on their knowledge of and understanding of external structures to engage with the technology. Therefore, we aimed to examine the specific question: 'In what ways did external, internal and technological structures impact on the implementation and adoption of the EMOS?'

## METHODS
### Study design and setting
Our study used a qualitative design. The study setting was a CCG in the south of England, which was chosen because it was an early adopter of the EMOS and had all general practices signed up to the system. The CCG was relatively small in size (17 separate general practices, and approximately 140 000 patients). Medicines management activities at the CCG were undertaken by three clinical pharmacists (including participants CCGP1 and CCGP2) and two pharmacy technicians. Additionally, one GP (participant GP1) operated as prescribing lead for the CCG. In the English National Health Service (NHS), a CCG is a clinically led statutory NHS body responsible for the planning and commissioning of healthcare services for their local area and cover groups of general practices within that area. The sampling frame was people within the CCG's geographical area who represented the stakeholder groups. This included doctors, pharmacists, general practice managers and patients.

### Understanding the background
Prior to data collection, we undertook actions to build a picture of the system and the context in which it was to be used. Authors MJ and RLH were given an overview of the system in a preliminary meeting with the study CCG prior to data collection. In addition, MJ visited a separate CCG in the north of England that was utilising the EMOS. Web-based materials relating to the system were read prior to data collection.[36]

### Recruitment and data collection
Individual participants were recruited on a purposive basis via the CCG or through community pharmacy networks to represent the different stakeholder groups (see table 1).

Potential participants were contacted by telephone or email. Five semi-structured interviews (lasting between 20 and 50 min) were conducted with three GPs and two CCG pharmacists, who were known to be using the system and had specific roles that required the use of the EMOS between August and December 2014. Four homogeneous focus groups (lasting between 57 and 112 min) were also conducted between September and December 2014, each with a specific group of stakeholders: GPs (2), community pharmacists (4), patients (4) and general practice managers (4). No repeat interviews were conducted, although one GP was interviewed and also participated

**Table 1** Case study participants

| Participants | Role | How they used the EMOS |
|---|---|---|
| **Interviews** | | |
| GP1-INT | General practitioner | In general practice and prescribing lead for the clinical commissioning group (CCG). Worked with the medicines management team in supporting the adoption of the EMOS by the CCG. Used the EMOS to send alerts to GPs. |
| GP2 | General practitioner | In general practice and respiratory lead for the CCG. Utilised the EMOS to undertake audits of prescribing relating to respiratory conditions. |
| GP3 | General practitioner | In general practice. |
| CCGP1 (additional observation as part of interview) | CCG pharmacist | Utilised the EMOS to undertake medication reviews with care home patients. |
| CCGP2 | CCG pharmacist | CCG medicines management team. Used the EMOS to run audits centrally at the CCG and then alert clinicians locally. |
| **Focus group A—general practitioners** | | |
| GP4 | General practitioner | In general practice. |
| GP1-FG | General practitioner | In practice and as prescribing lead for the CCG. |
| **Focus group B—community pharmacists** | | |
| CP1 | Community pharmacist | Aware of, but no access. |
| CP2 | Community pharmacist | Aware of, but no access. |
| CP3 | Community pharmacist | Aware of, but no access. |
| CP4 | Community pharmacist | Aware of, but no access. |
| **Focus group C—patients** | | |
| Pt1 | Patient | Access through patient passport. |
| Pt2 | Patient | Access through patient passport. |
| Pt3 | Patient | Access through patient passport. |
| Pt4 | Patient | Access through patient passport. |
| **Focus group D—general practice managers** | | |
| GPM1 | General practice manager | In general practice. |
| GPM2 | General practice manager | In general practice. |
| GPM3 | General practice manager | In general practice. |
| GPM4 | General practice manager | In general practice. |

CCG, clinical commissioning group; EMOS, electronic medicines optimisation system.

in a focus group. Each focus group was conducted with a different specific type of stakeholder, as this was felt to facilitate free and open discussion.

Topic guides for the interviews and focus groups were developed by reading relevant literature examining the implementation of IT in healthcare settings.[16 17 20 23 24] Both interviews and focus groups were conducted to illicit individual thoughts and opinions and to promote discussion among specific homogeneous groups of stakeholders. In the interviews and focus groups, we explored experiences of working with the EMOS, perceptions of the system, benefits and drawbacks, organisational structures and roles required for its use and the circumstances under which it was considered most effective. Data collection continued until saturation was reached, and no new themes emerged from the interviews and focus groups. The interviews and focus groups were carried out by a male research associate in medication safety trained and experienced in qualitative

health research who holds an MSc in Health Psychology (MJ). The focus groups were co-facilitated by a female freelance research pharmacist experienced in qualitative methodology and with a PhD in Medicines Safety in Primary Care (RLH). The researchers were not known to the participants prior to the study. Four interviews were conducted by telephone and one at the CCG offices. The focus groups were conducted at the CCG offices or at a local hotel solely with the participants, RLH and MJ present. All participants gave written informed consent to take part in the study and for the interviews and focus groups to be audio recorded and transcribed verbatim. Ethical approval for the study was granted by the Preston NHS Research Ethics Committee (reference 14/NW/0113).

### Analysis

The analysis was thematic, using a template approach.[37] Template analysis involves the summarising of themes

through a coding template. Often, template analysis begins with an a priori set of themes. New themes are then added, or existing themes revised, as data are iteratively analysed in a process of developing a template.[37] An a priori set of thematic codes based on SST was developed from the literature.[30 32 34 38] These included external structures such as national or local policies, guidelines and governance; interactions, including relationships, conflicts and communication; internal structures of agents, including dispositions, skills, attitudes and cognitive demands; rules and contextuality, including routines, social norms and regulations; and technological structures, including the social structures built into the technology. This set of codes was applied to the transcripts by MJ and documented using the QSR NVivo 10 application. The coding template was then modified through successive readings of the data and discussions with other authors. The template was then internally reviewed for completeness by MJ and DLP (who had independently reviewed all transcripts).

## RESULTS

The ways in which the EMOS was implemented and adopted were conceptualised in four broad thematic categories: adoption of the system for information gathering, perceptions of the system as new, perceptions of the EMOS as requiring technical competence and the interactions and relationships that involved individual or collective use of the technology.

### Adoption of the system for information gathering

The EMOS facilitated the efficient acquisition of information relating to the appropriateness of prescribing for individual patients. External structures provided the conditions for the use of the technology, specifically through the requirements of national policies relating to safe medicines use as set down by national governance and guidelines and the CCG's responses to those requirements. The CCG was motivated to carry out audits of prescribing, and much of the data extracted through such audits were used to benchmark the CCG against these national policies and targets. This auditing was in turn determined by the policy and institutional climate that required the reporting of such auditing, the setting of certain guidelines and targets and the adherence to those. This further led to the CCG utilising the technology in a local context to monitor prescribing behaviour in practices in response to local initiatives. External structures such as national or local 'initiatives' worked with the internal structures (in this specific instance, the motivations of the CCG to report in response to these 'initiatives') and the material properties of the technology, to more swiftly identify patients registered with general practices that met the relevant prescribing safety audit. The material properties of the system shaped the ability to conduct extensive searches of electronic health records across multiple general practices in a relatively short space of time. According to the

following extract from an interview with a CCG pharmacist, the technological structures enabled the collection of data in a more efficient and timely fashion.

> [...]it's a way of being able to gather pseudo-anonymised individual patient data and relate it to ideas and thoughts around initiatives that CCG or the medicines management team are looking at that perhaps has been identified or highlighted nationally, or locally and it can all be done relatively quickly within a few seconds if necessary. So you don't have to trawl round 17 different practices. (CCGP2)

Centrally, in a form of pay for performance initiative, the CCG made the EMOS part of a 'GP incentive scheme to engage with alerts in a meaningful way' (GP1-INT), and this was conceptualised as 'trying to sort of get some more traction' (GP1-INT). Guidelines and documents concerning strategies for prescribing framed the possibilities for use 'to actually monitor the progress against a sort of target outcome' (GP1-INT). The functionality within the EMOS allowed for benchmarking across the CCG. This in turn provided for structures that could be utilised by the CCG to encourage practices to use the system and an infrastructure that supported their own activities in monitoring prescribing behaviour and to 'reward good prescribing' (GP1-INT).

> 'if there are some practices that are demonstrating very good prescribing, then we've picked those out as well and highlighted those to act as a kind of beacon of hope for everybody else'. (CCGP2)

The system also allowed for communication channels and feedback, where contact with practices was through the system or as a result of alerts being sent out by email. Such communication, between the clinicians placed centrally at the CCG and the individual GP practices, enabled the CCG to monitor prescribing as 'a way of looking at the map' (GP2) as well as the use of the system by 'tracking our advice in those practices' (GP1-INT). The codes and material properties of the system facilitated monitoring in that logging on to the system indicated engagement with it. This in turn allowed the CCG to further monitor and audit prescribing patterns because they could swiftly see which practices had responded to alerts and '[could] have some kind of objective measure that [gave them] some idea as to who's perhaps even more engaged than others' (CCGP2). The ambitions and motivations of the CCG to monitor prescribing acted as an internal structure to work 'very hard to get the uptake of that better' (CCGP2) and in 'trying to persuade our clinicians to use it so that we get a much more real time feedback' (CCGP2). Furthermore, this combination of technological infrastructure and the ambitions of the CCG created a new internal structure in the form of a convention for using the system.

> (When) the GP logs onto the Eclipse system and there's a little tick box to say patient reviewed [...].

Now some practices are doing that as a regular routine exercise, so that means that tracking our advice in those practices is very easy and what it does allow you to do as well is not to send the same alert out to the same practice again. (GP1-INT)

In this way, there were patterns of agent–technology relationships that reinforced a hierarchical agent–agent relationship within the network. The CCG managers were interacting with the technology to monitor prescribing since engagement at local clinician level with the system was encouraged by the CCG. Because it provided further feedback to them, agent–technology relationships could build through the system use as new agent–agent relationships between managers centrally at the CCG and local GPs.

### Perceptions of the system as new
Using the EMOS was characterised as a new practice that would require new approaches. Resistance towards the system was thus justified by characterising existing behaviours as ingrained. Here, habits and ways of doing things that were presented by one GP as 'the old fashioned way' (CCGP2) provided for a limited use of the system. One such disposition was around their prescribing habits, which they described as 'conservative' (CCGP2). This allowed for a limited use of the EMOS, in which most alerts would not require action because prescribing behaviour was already 'protective of patients' (CCGP2). Similarly, as the following extract illustrates, non-use of the system resulted from habitual accustomed practice of using other systems, pre-existing routines and repetitive ways of doing things.

I think the trouble is Eclipse is another thing you have to log into along with the other 20 things you log into every day, and you're so used to using your other clinical system all the time. (GPM3)

In a further example of agent–agent relationships associated with the use of the system, the CCG pharmacists were concerned that GPs would otherwise avoid using the system. It was assumed that GPs, in addition to training on the system, needed persuasion in order to 'just [get] them to use it as habit' (CCGP1).

but we have had a situation where the GP said, oh, I'm not sure if I'll have time to look on Eclipse, but you can't spoon feed them everything. (CCGP1)

Social structures could shape the ways things were done. Workplace routines and practices, such as the prioritisation of work schedules, acted as constraints or enablers to the use of the new system. Here this GP highlighted contingencies within the structures associated with the 'special circumstances of my workplace' (GP4), which allowed for a range of actions from sidelining the alert through to reviewing the patient. In this way, the duality of structure—the specific demands of his work—and his agency—his interaction with the alerts in the EMOS—both

governed his act of utilising the system and the extent and character of that utilisation.

[…] it can depend on the nature of the alerts, how urgent it seems, and the special circumstances of my workplace […] some things might actually get side-lined for a few weeks if they're not clinically urgent, but […] the next time I catch up with my paperwork then I'll dig up that alert […] and review the situation. (GP4)

For the CCG pharmacist undertaking medication reviews in care homes, the system changed the way they worked because 'if necessary if there's something that comes up on Eclipse whilst we're there we can, rather than having to go back to the surgery first, check it and then make a decision' (CCGP1). In this way, the technology shaped their actions. Furthermore, the technological structures in the EMOS and the internal structures led to new shared decision making, use and outcome.

We can look on Eclipse and most of the time it's on Eclipse and we can answer the question there and then. For example, we had a patient who was on Memantine, who was a really not very well gentleman, […] so we phoned the GP straightaway. (CCGP1)

### Perceptions of the EMOS as requiring technical competence
The EMOS was conceptualised as a 'clever' system that could conduct complex searches but would require technical knowledge on the part of users in order to do so. This allowed for this GP's limited use of the system when combined with an understanding of his own abilities to use the system.

That's how I become accustomed to doing things, which is perhaps why I then don't use Eclipse, because I do think I might not have the ability and the power of making the use of a more powerful tool. But, perhaps I have also then learned useful habits with the old fashioned way. (GP4)

Non-use of the system was associated with the cognitive and physical demands associated with using the EMOS and finding time to learn how to get the best out of it. This further conceptualised the system as complex requiring time, training and 'proper teaching' (GPM2) to gain the expertise required to use it.

And if you had the time to log into it and go oh, what does this do? What does that do? […] You train your audit clerk who runs all sorts of searches and does all sorts of audit work, you could have the time to show her and teach her, […] I'd love to have the time to tinker with, (the system).[…] You'd need time to play with it and time to … proper teaching, proper (training) showing us what it does. (GPM2)

The conceptualisation of the EMOS as requiring technical competence was related to structures embedded within the technology that allowed for or constrained its

use. This could either empower users and thus facilitate further use or undermine that agency.

'And also I'm computer literate and I can work out, I can problem solve because I'm reasonably well educated, if you were talking about average population here, they would either give up, they would probably have given up when they couldn't log in'. (Pt2)

This service user conceptualises the system as difficult and one that required her abilities as a 'computer literate' to use it. This required an interaction of her capabilities and the structures within the system to engage with it, and difficulties with logging in was perceived to be a potential constraint for other users.

### Interactions and relationships: individual, shared and collective use of the technology

There were variations in the ways the technology was used within collaborative networks of social relations. Different general practice staff took responsibility for using the technology; use depended on shared or collective roles, or on a hierarchical allocation of access.

For service users, using the technology was determined by networks of social relations. This was expressed as having support from medical professionals to understand the system.

But I think the important thing is before you sort of almost start using it, you do need that kind of intervention from a medical practitioner in some way to actually help you with the things you need to know. (Pt1)

Within general practices, there was variation in who took responsibility for the EMOS. On receiving an alert through the system, one practice manager would then 'pass it on to the GP and get them to respond to me' (GPM 2) and that 'the doctors don't access it at all. […] I'm the only one that, yeah, has anything to do with it' (GPM 2). Another remarked,

I get the alert the same way through the email, I identify the patient […] then mine goes to the GP. But the GP actions it, I don't have any more responsibility for it after that […] They go into Eclipse, they do it, […] my job is just to literally give them the information and they do the rest. (GPM1)

Such variation was driven by the conventions and norms associated with work practices. In different general practices, individuals were assigned to different roles and responsibilities often based on what worked best for the practice.

one of the GPs has been nominated within our practice to take that lead in the same way that we break our workload down in other areas; you be the lead for this and tell us if there's anything we all need to know and share the workload. (GPM4)

The allocation of access to the EMOS limited its use. Community pharmacists did not have access to the system. Perceived social norms were seen as 'historically a barrier' (CP2) that perpetuated that lack of access. Community pharmacists attributed this barrier to GPs seeing themselves as 'as the custodians of the patient record' (CP2).

I think there always has been a conflict because GPs often see themselves as the custodians of the patient record and even though the information in that patient record, even abbreviated information is incredibly useful for community pharmacists, they've never successfully managed to allow us access and this is going back to EPS [Electronic Prescription Service], this is what EPS promised and it's never happened. (CP2)

There was a perception that the system was a tool for the CCG. This differential access meant that the system had not been used in some general practices. There were however perceptions that the system had 'evolved'.

I think that's what it was […] originally purchased … or the agreement with Eclipse was originally for the meds management team to use it as a tool for them […] And I think Eclipse has evolved since that happened […] And I don't think any of us have kept up with how Eclipse has evolved and what else it can now do. (GPM3)

Such changes were related to social norms around ownership and conventions concerning how the system would be used, centrally by the CCG to look at prescribing patterns across practices and by individual practices of their own prescribing audits. As the system evolved, there were perceptions that it could do more. In this way, perceptions of the technological structures and material properties of the technology drove the ambitions of some users to learn more about the potential uses of the system, which opened up access to different users.

### DISCUSSION

The adoption and implementation of the EMOS was dependent on a dynamic mix of external structures, internal structures and material properties embedded in the technology. External infrastructures, motivations of users and material properties of the EMOS facilitated information gathering. Perceiving the system as new could lead to resistance and maintenance of habitual behaviours. Use was dependent on interactions and relationships between users. Use could be further constrained by conceptualising the system as requiring technical competence.

SST proposes that in order to act, agents draw on internal structures. These internal structures include dispositions and knowledge of the 'strategic terrain' of external structures.[29] It has been suggested that to understand the implementation and adoption of IT from an SST standpoint, it is important to understand

the context in which the IT is being introduced, the networks of people and technologies, the dispositions of actors in those networks, the material properties of the technology and how these shape human action.[29 32] In the present study, the contextual background was shaped by policy relating to medication safety and the requirement to benchmark against national prescribing and safety targets. CCG managers' knowledge of the external structures relating to that policy background and their own skills and ambitions led to actions around the monitoring of prescribing behaviours across the CCG area. This was facilitated by material properties in the system. The outcomes of the monitoring actions were not just that prescribing data were gathered and reported to other institutions but that the external structures, the dispositions of the CCG managers and the material properties of the system allowed for governance and monitoring of clinician behaviours through tracking engagement with the system and processes of persuasion and reward. This could therefore have been said to reinforce hierarchical relationships between the CCG and the local GPs. Hence, the use of the system created new internal structures concerning such social rules and conventions. Similarly, in previous literature, information systems have been associated with enabling managers to capture information, place local clinicians under surveillance and make their actions calculable.[39] Furthermore, an effect of such surveillance is for individuals to adapt their own behaviour to ensure they act legitimately.[40]

Previous literature has established the role of social, organisational and work practices in the adoption of IT[16 17]; others have focused on functionality of design and tailoring to users[18 41] or on top-down implementation.[41 42] Other research has indicated that emphasis on training might also construct end users as the problem.[43] With notable exceptions,[29 33 34] much of this earlier literature has highlighted the importance of work practices and how technology needs to be embedded into pre-existing routines but has not seen these dynamically linked to wider contexts particularly in the context of medication safety in primary care. In this study, we found that key agents in the network either resisted or sustained use of the system. GPs saw the system as new and unnecessary and not compatible with existing workplace routines. There were also differences in agents' responses to the material properties in the system where these were seen as facilitating use by some agents and by others as a barrier to use because the material properties of the system were perceived as to make it to difficult to use. SST enabled us to understand these dispositional behaviours in relation to social structures, particularly pre-existing routines, work practices and social norms. In previous research, there has been a focus on interoperability, work practices and system usability, suggesting that poor adoption of IT is related to users or the system.[19 41] This misses how interactions and relationships between contexts, users and technology might work and how the implementation of

IT is a social practice.[39] In this study, these networks of social relations affected the use of the system.

## Implications of the findings

SST would argue that individual agency is dependent on knowledge of rules and conventions. Primary care settings are governed by institutional norms, measures, rules and traditions, habits and behaviours.[29 44] Some of these are embedded in local rules and conventions associated with the different working dynamics of individual practices, whereas others are found in regulations and governance associated with wider economic and institutional contexts.[29 45] Using SST in this way may be particularly valuable in primary care research because general practices operate with their own organisational culture and dynamic, which may well lead to marked differences in working practices and structure.[46]

This study highlights how healthcare IT interventions are implemented and adopted in a complex social and organisational context. Interventions that are top-down and perceived as tools of managerial control are less likely to be effective than those that take into consideration existing local practices and the ambitions and attitudes of those who will use the technology.

## STRENGTHS AND LIMITATIONS

Much of the previous literature on interventions to improve medication safety has focused on secondary healthcare settings, and electronic audit and feedback systems of the kind examined in this study are under-researched in primary care. A particular strength of this study is the use of SST, which was found to be a useful theoretical approach to studying the implementation and adoption of the EMOS in a primary care setting. In applying this theoretical approach, we were able to see the differences in motivations, ambitions, aims and attitudes of different actors from different stakeholder groups towards the IT intervention. SST could also reveal the complex contextual background in which the EMOS was implemented, and it revealed how the implementation was informed by wider contexts. Hence, we were able to understand that the successful adoption of the EMOS was not merely dependent on agents but on the complex terrain in which it was implemented. Previous studies using this approach have focused on large national IT projects where institutional contexts might be considered to have more effect.[32 47] We found, however, that in a smaller-scale project, wider policy institutional contexts did affect the implementation and adoption of the IT, for example, through the CCG's response to the requirements of national policies. In this way, the use of the system depended on other factors alongside the dispositions of the users.

There are several limitations to this work, which present further opportunities for future research examining the adoption and implementation of electronic audit and feedback systems to improve medication safety in primary

care settings. It has been suggested that studies such as these explore wider social contexts through analysis of background data and through ethnographic observation.[35 48] Although we conducted one observation with a CCG pharmacist, this was only as an extension of the interview with that participant to elicit some further understanding of how they used the EMOS. A number of naturalistic observations would have been useful in unpicking contexts and agents' choices and actions in using the system.

## CONCLUSION

Our study examines the implementation and adoption of an IT system for medicine optimisation in primary care. It was found that the dynamic combination of external, internal and technological structures affected the adoption and implementation of the system. IT interventions for medicine optimisation should consider how utilisation may depend on a combination of the infrastructure 'within' primary care, the social structures embedded in the technology and the conventions, norms and dispositions of those utilising it.

**Acknowledgements** The authors are grateful to all participants who kindly gave their time. They would also like to acknowledge the assistance given by the NIHR Clinical Research Network.

**Contributors** All authors were involved in the design of this study. MJ led on recruitment of participants, data collection, analysis of the data and drafting of the article. RLH helped co-facilitated focus groups. DP, RLH. SR, AA and DA also made contributions to analysis and interpretation. All authors revised the article critically and approved the final version to be published.

**Funding** This research was funded by the National Institute for Health Research through the Greater Manchester Primary Care Patient Safety Translational Research Centre (NIHR GM PSTRC), grant number gmpstrc-2012-1.

**Competing interests** None declared.

**Ethics approval** All participants gave informed written consent to take part in the study, and for the interviews and focus groups to be audio recorded and transcribed verbatim.

**Ethics approval** Ethical approval for the study was granted by the NHS National Research Ethics Service (reference 14/NW/0113).

**Provenance and peer review** Not commissioned; externally peer reviewed.

**Data sharing statement** Data cannot be shared because participants did not consent to this. In addition, because this is a small case study involving small numbers of participants, there is a possibility that material in the transcripts could identify participants.

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
