## [Reviewer comments · BMJ Open]

ARTICLE DETAILS

TITLE (PROVISIONAL)	Understanding the implementation and adoption of an information technology intervention to support medicines optimisation in primary care: qualitative study using strong structuration theory
AUTHORS	Jeffries, Mark; Phipps, Denham; Howard, Rachel; Avery, Anthony; Rodgers, Sarah; Ashcroft, Darren

VERSION 1 - REVIEW

REVIEWER	Trisha Greenhalgh U of Oxford, UK
REVIEW RETURNED	23-Nov-2016

GENERAL COMMENTS	Broadly, a good application of SST to a very interesting study. Three things that need more work (I've put 'major revision' because they do need a bit of thinking through, but I think the authors are up to this task. 1. Need to explain '[social] structures'. BMJ Open readers are (mostly) a sociologically naive audience. The current explanation goes too fast, assumes too much prior understanding. You're using 'structures' in many different ways in the paper. Only some of the uses relate to SOCIAL structures. All this needs unpacking, explaining and extending. Ditto infrastructures.2. Remove reference to technologies as 'equal partners'. Rob Stones (who developed SST and worked with me to adapt it to technology programmes) is adamant about this. It's ANT that insists on 'symmetry' between human and tech. Tech, as we said in our Soc Sci Med paper, can only ever be an 'actor' in the network to a limited and contingent degree.3. Discussion is too short and under theorised. This is partly because a significant limitation is you only had interview / focus group data. This limitation should be acknowledged and discussed. You never actually studied directly what people did. I also think you should go back and look at policy documents, the absence of which is a limitation you could address. Your qualitative data is lovely but it's only one layer of a multi-layered system.4. I don't like Figure 1, for two reasons. One is you are using deterministic language, which seems naive. The other is you have made SST into a flow chart with technology structures as (it seems) an intervening variable. I agree the figure Rob Stones and I created (ref 26) is somewhat more complex and may need simplifying (though if you'd like to use it, feel free - would need to ask Soc Sci Med) but I am very worried that this new diagram is going to depict an inaccurate version of the theory which may then be further
---

	distorted as people interpret it as a deterministic flowchart. All the above could be rectified, happy to look at a revision.
--	---

REVIEWER	Albert Farre University of Birmingham, UK
REVIEW RETURNED	14-Dec-2016

GENERAL COMMENTS	Thank you for the opportunity to review this paper, which explores an important and topical issue through relevant methodological and theoretical tools. The paper is clearly written and well structured. In my opinion, the main core issues that need addressing relate to the methodology and the particular use of Strong Structuration Theory (SST) in this study: Although the aim of the research is stated at the end of the introduction, it would be useful for the reader to know what the specific research question(s) was. Particularly, given that SST is part of the formulation of the aim, it would be relevant to know if SST informed the specific research question(s) addressed. Did SST inform the development of the topic guides for the semi-structured interviews? If so, this should be reported. If not, a general indication as to how the topic guides were developed would increase the quality of the reporting. Also, further clarification should be provided about key elements of SST in this particular study and whether the methods used were well suited to address these:  • How were the material properties of the technology conceptualised/mapped out? Was that there any data collected concerning the material properties of the technology? If not, why not and what assumptions were made in order to use SST to think through the implementation/adoption problem explored? • How was the institutional context explored when mapping the network-in-focus? • The findings seem to focus on agent–technology relationships, but how were the agent–agent and technology–technology relationships in the network explored? • Was ‘change over time’ captured? If not, why was this omitted and how was the ‘stability’ of the network judged? • The findings and the discussion do not seem to refer to how the social structures inscribed in the technology enable, influence, or constrain the active agency of users – despite this having been highlighted as one of the key features/contributions of SST. The appropriateness of the research design to address the aim of the study should be further justified, particularly in light of (and in relation to) the choice of a case study design. Within this, there is a core tension that needs to be addressed, given that case study research focuses on providing an in-depth understanding of the unit of analysis (usually through multiple methods), the authors should
---

	clarify the rationale behind pairing such a design with a single-method (individual/group interviews) data collection process and a deductive approach to data analysis. In addition, the paper would be strengthened if the authors reported what was the particular approach to case study research that informed their methodological decision-making. The sampling strategy should be further clarified – what were the authors aiming for? Why some stakeholders are only interviewed individually or in group? What was the rationale for the size and composition of the groups, which seemed to favour small and homogeneous groups? Although the authors claim that saturation was achieved (Page 7, Line 34), it should be further clarified how this was achieved in the context of a deductive approach to qualitative data analysis where thematic codes were established a priori (Page 8, Line 6). Tied to this, 'emerging' themes are still mentioned in both pages and is therefore unclear what the actual approach to data analysis was. Thus, it is paramount for the reader to know a bit more about the rationale and process followed for data analysis. In doing so, the authors may want to consider reporting any procedures they may have put in place to ensure the credibility/trustworthiness of the findings.
--	---

VERSION 1 – AUTHOR RESPONSE

Reviewer: 1

Reviewer Name: Trisha Greenhalgh

1. Need to explain '[social] structures'. BMJ Open readers are (mostly) a sociologically naive audience. The current explanation goes too fast, assumes too much prior understanding. You're using 'structures' in many different ways in the paper. Only some of the uses relate to SOCIAL structures. All this needs unpacking, explaining and extending. Ditto infrastructures.

Thank you for highlighting this. We accept that the ways we have explained SST and structures is currently an incomplete explanation. We have rewritten the paragraph starting page 6 line 8 which now reads:

“Strong structuration theory (SST) has been proposed as a way of examining these sociotechnical aspects of healthcare IT implementation [30]. It is based on Giddens' structuration theory, which proposed a relationship between structures (such as social norms, political and economic institutions) and agency (people's actions and choices) [31]. According to Stones [30], SST extends this structure-agency relationship to include the following elements (see figure 1):

- External structures, which are the physical, social or economic context in which action is contemplated. External structures are built through social positions, practices and networks of social relationships [29, 32]. These could include hierarchical relationships between employers and employees, professional roles, local and national guidelines, governance measures, regulations, professional codes of practice, as well as local work practices and interactions among groups of stakeholders [30, 32].
- Internal structures, which are manifest in two ways. Firstly, as the skills, dispositions, ambitions, attitudes, values, past experiences of actors and ways of viewing the world. Secondly, as the actors' knowledge of rules, conventions, obligations and social norms, which may involve partial understandings and past experiences. These inform how one is supposed to act in specific situations in the here and now, based upon the agents understanding of external structures [30, 32-33].
- Agency, which is how and why agents draw upon internal structures to act in particular ways in specific situations [32]

- Outcome, which is the way agency impacts on external or internal structures and how they are maintained or changed [32].

Stones and Greenhalgh [32] further explained the role of technology in SST: rather than there being symmetry between technology and human actors, they are instead separate and may act in different ways [32]. Technology incorporates procedures, codes, material properties and standards that can enable or constrain use [19, 32-34]; it is therefore seen as shaping human actions by making certain actions possible [35]. Previous studies [29,34] suggest that SST can illuminate the implementation and adoption of information technology by understanding how people "take action with respect to technologies"; in other words, what people actually do with the systems and to what effect [32]. Strong structuration theory has been previously used to understand the ways a large scale healthcare IT intervention, designed to assist patients and GPs to book hospital outpatient appointments, was resisted or adopted [34]."

On reflection we feel that the use of the term infrastructures in the paper may have been misleading. By using this we were attempting to highlight those structures associated with governance, policy and guidelines. We now think it better to be more specific in how we frame this so have made a number of changes throughout the manuscript. We have in addition made a number of small changes to the wording elsewhere in the manuscript to improve the clarity, readability and accessibility of the paper for the particular audience of BMJ Open.

2. Remove reference to technologies as 'equal partners'. Rob Stones (who developed SST and worked with me to adapt it to technology programmes) is adamant about this. It's ANT that insists on 'symmetry' between human and tech. Tech, as we said in our Soc Sci Med paper, can only ever be an 'actor' in the network to a limited and contingent degree.

We completely accept this point and have changed the text accordingly as noted in the previous response.

3. Discussion is too short and under theorised. This is partly because a significant limitation is you only had interview / focus group data. This limitation should be acknowledged and discussed. You never actually studied directly what people did. I also think you should go back and look at policy documents, the absence of which is a limitation you could address. Your qualitative data is lovely but it's only one layer of a multi-layered system.

We agree that the reliance upon interview/focus group data is a limitation. We now acknowledge this more clearly at the start of the paper on page 3 in the "Strengths and Limitations section".

We have looked at policy documents, relevant to the time period of data collection that related to medication safety, patient safety and the utilisation of information technology. We have included new references to these and added this paragraph to the Introduction at page 4 line 13, to provide a contextual background relating to relevant policy.

"In the UK, national and local policies have set out recommendations for medication safety improvement [10-12]. "An Organisation with a Memory" [13] set out the necessity for the establishment of a patient safety culture within healthcare organisations. This emphasised the importance of organisational practices. Policy has also set out how the utilisation of information technology presents opportunities to fulfil medication safety requirements and that the contribution of information systems should be maximised [11]. This was further enhanced by Department of Health recommendations in the report "Building a Safer NHS for patients: Improving Medication Safety" [10], which recommended that steps to safer prescribing may include the implementation of effective IT systems particularly those systems that might highlight and give warnings to medical staff of prescription errors. Similar recommendations have suggested there is a need to develop systems that optimise the use of medicines and that this might include improved electronic decision support for clinicians [12]. Locally, the Clinical Commissioning Group (CCG) that formed the setting for this study operated a prescribing incentive scheme designed to improve the quality of prescribing, respond to the requirements of national guidelines, and reduce excessive prescribing and costs, which was incentivised by small financial rewards for general practices [14]."

We have made substantial changes to the discussion. This from page 19 line 8 to page 21 line 22

now reads:

“The adoption and implementation of the EMOS was dependent upon a dynamic mix of external structures, internal structures and the material properties embedded in the technology. External infrastructures, the motivations of users and the material properties of the EMOS facilitated information gathering. Perceiving the system as new could lead to resistance and the maintenance of habitual behaviours. Use was dependent upon interactions and relationships between users. Use could be further constrained by conceptualising the system as requiring technical competence. SST proposes that in order to act, agents draw upon internal structures. These internal structures include dispositions and knowledge of the "strategic terrain" of external structures [29]. It has been suggested that to understand the implementation and adoption of IT from a SST standpoint it is important to understand the context in which the IT is being introduced, the networks of people and technologies, the dispositions of actors in those networks, the material properties of the technology and how those shape human action [29, 32]. In the present study, the contextual background was shaped by policy relating to medication safety and the requirement to benchmark against national prescribing and safety targets. CCG managers' knowledge of the external structures relating to that policy background, and their own skills and ambitions, led to actions around the monitoring of prescribing behaviours across the CCG area. This was facilitated by material properties in the system. The outcomes of the monitoring actions were not just that prescribing data was gathered and reported to other institutions but that the external structures, the dispositions of the CCG managers and the material properties of the system allowed for governance and monitoring of clinicians behaviours through tracking engagement with the system and processes of persuasion and reward. This could therefore have been said to reinforce hierarchical relationships between the CCG and local GPs. Hence, the use of the system created new internal structures concerning such social rules and conventions. Similarly, in previous literature, information systems have been associated with enabling managers to capture information, place local clinicians under surveillance and make their actions calculable [39]. Furthermore, an effect of such surveillance is for individuals to adapt their own behaviour to ensure they act legitimately [40].

Previous literature has established the role of social, organisational and work practices in the adoption of IT, [16,1] others have focused upon functionality of design and tailoring to users [18, 41] or upon top-down implementation [41, 42]. Other research has indicated that emphasis upon training might also construct end-users as the problem [43]. With notable exceptions [29, 33, 34], much of this earlier literature has highlighted the importance of work practices and how technology needs to be embedded into pre-existing routines, but has not seen these dynamically linked to wider contexts particularly in the context of medication safety in primary care. In this study we found that key agents in the network either resisted or sustained use of the system. GPs saw the system as new and unnecessary and not compatible with existing workplace routines. There were also differences in agents responses to the material properties in the system where these were seen as facilitating use by some agents and by others as a barrier to use because the material properties of the system were perceived as to make it difficult to use. SST enabled us to understand these dispositional behaviours in relation to social structures particularly pre-existing routines, work practices and social norms. In previous research there has been a focus upon interoperability, work practices and system usability suggesting that poor adoption of IT is related to users or the system [19, 41]. This misses how interactions and relationships between contexts, users and the technology might work and how the implementation of IT is a social practice [39]. In this study, these networks of social relations impacted upon the use of the system.

Implications of the findings

SST would argue that individual agency is dependent upon knowledge of rules and conventions. Primary care settings are governed by institutional norms, measures, rules and traditions, habits and behaviours [29, 44]. Some of these are embedded in local rules and conventions associated with the different working dynamics of individual practices, while others are found in regulations and governance associated with wider economic and institutional contexts [29, 45]. Using SST in this way may be particularly valuable in primary care research, as general practices operate with their own

organizational culture and dynamic which may well lead to marked differences in working practices and structure [46].”

4. I don't like Figure 1, for two reasons. One is you are using deterministic language, which seems naive. The other is you have made SST into a flow chart with technology structures as (it seems) an intervening variable. I agree the figure Rob Stones and I created (ref 26) is somewhat more complex and may need simplifying (though if you'd like to use it, feel free - would need to ask Soc Sci Med) but I am very worried that this new diagram is going to depict an inaccurate version of the theory which may then be further distorted as people interpret it as a deterministic flowchart.

We have removed Figure one and replaced with that from ref 26 (now reference 32). This new figure is placed at the end of the Introduction. We agree that this is more appropriate. Approval for use of the figure and appropriate citation has been granted by Elsevier (licence number 4032471377107).

Reviewer: 2

Reviewer Name: Albert Farre

In my opinion, the main core issues that need addressing relate to the methodology and the particular use of Strong Structuration Theory (SST) in this study:

1. Although the aim of the research is stated at the end of the introduction, it would be useful for the reader to know what the specific research question(s) was. Particularly, given that SST is part of the formulation of the aim, it would be relevant to know if SST informed the specific research question(s) addressed.

We accept that stating the research aim without a specific research question might have been somewhat ambiguous. We have included the following question at the end of the introduction on page 8 line 6:

"Therefore, we aimed to examine the specific question: "in what ways did external, internal and technological structures impact upon the implementation and adoption of the EMOS?"

2. Did SST inform the development of the topic guides for the semi-structured interviews? If so, this should be reported. If not, a general indication as to how the topic guides were developed would increase the quality of the reporting.

The development of the topic guides was informed by relevant literature. We have added the following to the methods section at page 10 lines 10-11

"Topic guides for the interviews and focus groups were developed by reading relevant literature examining the implementation of information technology in healthcare settings [16, 17, 20, 23, 24.]"

3. Also, further clarification should be provided about key elements of SST in this particular study and whether the methods used were well suited to address these:

a. How were the material properties of the technology conceptualised/mapped out? Was that there any data collected concerning the material properties of the technology? If not, why not and what assumptions were made in order to use SST to think through the implementation/adoption problem explored?

Authors MJ and RLH were given an overview of the system in a preliminary meeting with the CCG prior to data collection. Web-based materials relating to the company were read prior to data collection. We have added further explanation about the system and a new reference (Eclipse Solutions Eclipse Live <https://www.eclipsesolutions.org/EclipseInfo/AboutEclipse/AccessedAugust2014>) at the end of the introduction section on page 7 line 16. This now reads:-

"The EMOS allows different stakeholders - general practitioners, Clinical Commissioning Group (CCG) managers, pharmacists, general practice managers, and patients - access to real time anonymized patient data including medical diagnoses, prescribed medications and laboratory test

results. It comprises a secure patient database and a web-based user interface that extracts patient specific data from the general practice clinical record system. The interface provides a number of user functions; these include reviewing a specific patient health record, identifying patients who are at risk of a medication-related adverse event, such as those who are on inappropriate combinations of drugs or who have not received appropriate monitoring and carrying out clinical audits on a subset of patients [36]. The EMOS also allowed clinicians and managers in the health locality to audit prescribing practices across general practices and make comparisons against national guidelines. Patients have access to the system through a patient passport which allows them to view their medications and test results."

In addition we have added this sentence on page 8 line 3 explaining further why SST was appropriate to explore the implementation and adoption of the system

"In this context it was felt that SST would unpick the ways in which users of the system drew upon their dispositions, attitudes skills and ambitions and upon their knowledge of and understanding of external structures to engage with the technology."

For further clarification we have added the following at the start of the methods in a new subsection subtitled "Understanding the background" at page 8 line 26 to Page 9 line 5.

This now reads:-

"Prior to data collection we undertook actions to build a picture of the system and the context in which it was to be used. Authors MJ and RLH were given an overview of the system in a preliminary meeting with the study CCG prior to data collection. In addition MJ visited a separate CCG in the North of England that was utilising the EMOS. Web-based materials relating to the system were read prior to data collection [36]."

b. How was the institutional context explored when mapping the network-in-focus?

The institutional context was explored through the activities described in 3a above, the interviews and the focus groups. Of particular use were the interviews with GP1 and CCGP2.

We agree that the nature of this institutional context requires some clarification we have added the following to the beginning of the Methods section page 8 lines 16-20 which now reads:-

"The CCG was relatively small in size (17 separate general practices, and approximately 140,000 patients). Medicines management activities at the CCG were undertaken by three clinical pharmacists (including participants CCGP1 and CCGP2) and two pharmacy technicians. Additionally one GP (participant GP1) operated as prescribing lead for the CCG."

c. The findings seem to focus on agent–technology relationships, but how were the agent–agent and technology–technology relationships in the network explored?

We feel that there are references to agent-agent interactions in the results. We respectfully refer to page 13 lines 21-25, page 13 lines 26 to page 14 line 12 and to the section of the results subtitled Interactions and relationships. We do however accept that this may well not be clear and have made changes in the results.

At page 14 lines 13-18 we have added this to further explain the network of relationships:-

"In this way there were patterns of agent-technology relationships that reinforced a hierarchical agent - agent relationship within the network. The CCG managers were interacting with the technology to monitor prescribing since engagement at local clinician level with the system was encouraged by the CCG, because it provided further feedback to them, agent-technology relationships could build through the system use as new agent -agent relationships between managers centrally at the CCG and local GPs."

At page 15 lines 6-9 now reads:

"In a further example of agent-agent relationships associated with the use of the system, the CCG pharmacists were concerned that GPs would otherwise avoid using the system. It was assumed that

GPs, in addition to training on the system, needed persuasion in order to “just [get] them to use it as habit” (CCGP1).”

We also thank you for raising this because it has highlighted how the networks associated with the use of the system could be clarified further. We have also made changes to the discussion to rectify this – (please see the response to reviewer 1 point 3 above).

d. Was ‘change over time’ captured? If not, why was this omitted and how was the ‘stability’ of the network judged?

It was not possible to capture change over time owing to the restricted time window for data collection between August and December 2014.

e. The findings and the discussion do not seem to refer to how the social structures inscribed in the technology enable, influence, or constrain the active agency of users – despite this having been highlighted as one of the key features/contributions of SST.

We respectfully draw your attention to page 12 line 26 to page 13 line 4; page 13 line 14-to page 14 line 12; page 15 line 23 to Page 16 line 5 -25 and page 17 lines 1 to 9. We accept that this is not drawn out in the discussion and have made further changes in this section - please see response to reviewer 1 point 3 above.

4. The appropriateness of the research design to address the aim of the study should be further justified, particularly in light of (and in relation to) the choice of a case study design. Within this, there is a core tension that needs to be addressed, given that case study research focuses on providing an in-depth understanding of the unit of analysis (usually through multiple methods), the authors should clarify the rationale behind pairing such a design with a single-method (individual/group interviews) data collection process and a deductive approach to data analysis. In addition, the paper would be strengthened if the authors reported what was the particular approach to case study research that informed their methodological decision-making.

We agree that this is unclear. We felt that this was a case study because of the specific contextual and geographical boundary of the single CCG utilising the system. We accept however that this in itself does not necessarily constitute a case study in that the methodological approach does not have multiple perspectives. We have therefore removed references to case study throughout the manuscript and refer instead to this as a qualitative study.

5. The sampling strategy should be further clarified – what were the authors aiming for? Why some stakeholders are only interviewed individually or in group? What was the rationale for the size and composition of the groups, which seemed to favour small and homogeneous groups?

Thank you for this comment and we agree that the sampling strategy requires further clarification. We choose to conduct both interviews and focus groups because through the interviews we were trying to find individual opinions, thoughts and understandings of the system from specific individual users.

Table 1 highlights that a number of the interview participants had specific roles that required them to use the EMOS in particular ways. The focus groups were intended to promote discussion amongst a specific homogenous group of stakeholders. It was felt that this approach would be more likely to illicit views than a mixed focus group. For instance the patient focus group could talk more freely (and critically) without the presence of health professionals. We felt that the focus groups were an appropriate size to facilitate rich in depth discussion in the time allowed. For clarification we have made some changes to Table 1 on page 9 and to the method section starting at page 7.

Table one now reads:-

Participants	Role	How they used the EMOS
--------------	------	------------------------

Interviews		
------------	--	--

GP1-INT	General Practitioner	In general practice and prescribing lead for the Clinical Commissioning Group (CCG). Worked with the medicines management team in supporting the adoption of the EMOS by the CCG. Used the EMOS to send alerts to GPs
---------	----------------------	---

GP2 General Practitioner In general practice and respiratory lead for the CCG. Utilised the EMOS to undertake audits of prescribing relating to respiratory conditions.

GP3 General Practitioner In general practice

CCGP1 (additional observation as part of interview) CCG Pharmacist Utilised the EMOs to undertake medication reviews with care home patients

CCGP2 CCG Pharmacist CCG medicines management team. Used the EMOS to run audits centrally at the CCG and then alert clinicians locally

Focus group A - General Practitioners

GP4 General Practitioner In general practice

GP1-FG General Practitioner In practice and prescribing lead for the CCG

Focus group B – Community Pharmacists

CP1 Community Pharmacist Aware of, but no access

CP2 Community Pharmacist Aware of, but no access

CP3 Community Pharmacist Aware of, but no access

CP4 Community Pharmacist Aware of, but no access

Focus Group C – Patients

Pt1 Patient Access through patient passport

Pt2 Patient Access through patient passport

Pt3 Patient Access through patient passport

Pt4 Patient Access through patient passport

Focus Group D - General practice managers

GPM1 General Practice Manager In general practice

GPM2 General Practice Manager In general practice

GPM3 General Practice Manager In general practice

GPM4 General Practice Manager In general practice

Page 10 lines 2-14 now reads:-

“Potential participants were contacted by telephone or email. Five semi-structured interviews (lasting between 20-50 minutes) were conducted with three GPs and two CCG pharmacists, who were known to be using the system, and had specific roles that required the use of the EMOS between August and December 2014. Four homogeneous focus groups (lasting between 57-112 minutes) were also conducted between September and December 2014, each with a specific group of stakeholders: GPs (2); community pharmacists (4); patients (4); and general practice managers (4). No repeat interviews were conducted, although one GP was interviewed and also participated in a focus group. Each focus group was conducted with different a specific type of stakeholder, as this was felt to facilitate free and open discussion.

Topic guides for the interviews and focus groups were developed by reading relevant literature examining the implementation of information technology in healthcare settings [16, 17, 20, 23, 24.] Both interviews and focus groups were conducted to illicit individual thoughts and opinions and to promote discussion amongst specific homogenous groups of stakeholders.”

6. Although the authors claim that saturation was achieved (Page 7, Line 34), it should be further clarified how this was achieved in the context of a deductive approach to qualitative data analysis where thematic codes were established a priori (Page 8, Line 6). Tied to this, ‘emerging’ themes are still mentioned in both pages and is therefore unclear what the actual approach to data analysis was. Thus, it is paramount for the reader to know a bit more about the rationale and process followed for data analysis. In doing so, the authors may want to consider reporting any procedures they may have put in place to ensure the credibility/trustworthiness of the findings.

We agree that this is somewhat unclear. Our approach to the analysis was consistent with template analysis which starts with a priori template of codes but then adapts, modifies and develops these in an inductive process throughout the analysis. We developed an a priori set of thematic codes as described in lines 13-14 on page 11. We did not however wish to be completely restricted to this

deductive approach so coded extracts were then read and re-read and discussed between authors. In this way new themes emerged and existing themes were expanded or collapsed. These new themes were then used to develop a new coding framework which was then applied to the transcripts. For greater clarity we have rephrased the section subtitled analysis (page 11 from line 10) which now reads:-

"The analysis was thematic, using a template approach [37]. Template analysis involves the summarising of themes through a coding template. Often, template analysis begins with an a priori set of themes. New themes are then added, or existing themes revised, as data is iteratively analysed in a process of developing a template [37]. An a priori set of thematic codes based upon strong structuration theory was developed from the literature [30, 32, 34, 38.] These included: external structures such as national or local policies, guidelines and governance; interactions, including relationships, conflicts and communication; the internal structures of agents including dispositions, skills, attitudes and cognitive demands; rules and contextuality including routines, social norms and regulations and technological structures including the social structures built into the technology. This set of codes was applied to the transcripts by MJ and documented using the QSR NVivo 10 application. The coding template was then modified through successive readings and of the data and discussions with other authors. The template was then internally reviewed for completeness by MJ and DLP (who had independently reviewed all transcripts)."

We have made some additional minor changes throughout the paper to improve the readability. Having made these changes to the paper we feel it is much stronger and has much greater clarity. We hope this is now suitable for publication in BMJ Open

VERSION 2 – REVIEW

REVIEWER	Trish Greenhalgh U of Oxford, UK
REVIEW RETURNED	31-Jan-2017

GENERAL COMMENTS	Much improved and they have responded well to the comments. One final quibble: the abstract needs more detail on methods. "transcripts were analysed thematically" doesn't cut the mustard!
---

REVIEWER	Albert Farre University of Birmingham, UK
REVIEW RETURNED	20-Feb-2017

GENERAL COMMENTS	Very interesting paper as stated before. I think the reporting is clearly improved after the revision and I would recommend publication.
--

VERSION 2 – AUTHOR RESPONSE

Reviewer One:-the abstract needs more detail on methods. "transcripts were analysed thematically" doesn't cut the mustard!

We have made changes to the "Design" subsection of the abstract. This now reads:-

A qualitative study informed by strong structuration theory. The analysis was thematic, using a template approach. An a priori set of thematic codes, based upon strong structuration theory, was developed from the literature and applied to the transcripts. The coding template was then modified through successive readings of the data.

We would like to thank the reviewers for their very detailed comments on the earlier version of this manuscript. We trust that this manuscript is now acceptable for publication in BMJ OPEN

VERSION 3 – REVIEW

REVIEWER	Albert Farre University of Birmingham, UK
REVIEW RETURNED	31-Mar-2017

GENERAL COMMENTS	In my opinion the authors have appropriately addressed the reviewers' comments.
---